# Evaluating the simulated mean soil carbon transit times by Earth System Models using observations

Jing Wang[1], Jianyang Xia[1,2*], Xuhui Zhou[1,2], Kun Huang[1], Jian Zhou[1], Yuanyuan Huang[3], Lifen Jiang[4], Xia Xu[5], Junyi Liang[6], Ying-Ping Wang[7], Xiaoli Cheng[8], Yiqi Luo[4,9]

[1]Zhejiang Tiantong Forest Ecosystem National Observation and Research Station, Shanghai Key Lab for Urban Ecological Processes and Eco-Restoration, School of Ecological and Environmental Sciences, East China Normal University, Shanghai 200241

[2]State Key Laboratory of Estuarine and Coastal Research, Research Center for Global Change and Ecological Forecasting, East China Normal University, Shanghai 200241, China

[3]Laboratoire des Sciences du Climat et de l'Environnement, 91191 Gif-sur-Yvette, France

[4]Center for ecosystem science and society, Northern Arizona University, Arizona, Flagstaff, AZ 86011, USA

[5]College of Biology and the Environment, Nanjing Forestry University, Nanjing 210037, China

[6]Environmental Sciences Division & Climate Change Science Institute, Oak Ridge National Laboratory, Oak Ridge, Tennessee 37830, USA

[7]CSIRO Ocean and Atmosphere, PMB #1, Aspendale, Victoria 3195, Australia

[8]Wuhan Botanical Garden, Chinese Academy of Sciences, Wuhan 430074, Hubei Province, China

[9]Department of Earth System Science, Tsinghua University, Beijing 100084, China

Keywords: CMIP5, global land model, model uncertainty, soil carbon, transit time, turnover time

*Corresponding author: Dr. Jianyang Xia

ORCID: Jianyang Xia: https://orcid.org/0000-0001-5923-6665

Tel.: + 86 021-5434 2677

E-mail: jyxia@des.ecnu.edu.cn;

**Abstract**

One known bias in current Earth System Models (ESMs) is the underestimation of global mean soil carbon (C) transit time ($\tau_{soil}$), which quantifies the age of the C atoms at the time they leave the soil. However, it remains unclear where such underestimations are located globally. Here, we constructed a global database of measured $\tau_{soil}$ across 187 sites to evaluated results from twelve ESMs. The observations showed that the estimated $\tau_{soil}$ was dramatically shorter from the soil incubations studies in the laboratory environment (Median = 4 years; interquartile range = 1 to 25 years) than that derived from field *in-situ* measurements (31; 5 to 84 years) with the shifts of stable isotopic C ($^{13}C$) or the *stock-over-flux* approach. In comparison with the field observations, the multi-model ensemble simulated a shorter median (19 years) and a smaller spatial variation (6 to 29 years) of $\tau_{soil}$ across the same site locations. We then found a significant and negative linear correlation between the *in-situ* measured $\tau_{soil}$ and mean annual air temperature. The underestimations of modeled $\tau_{soil}$ are mainly located in cold and dry biomes, especially tundra and desert. Furthermore, we showed that one ESM (i.e., CESM) has improved its $\tau_{soil}$ estimate by incorporation of the soil vertical profile. These findings indicate that the spatial variation of $\tau_{soil}$ is a useful benchmark for ESMs, and we recommend more observations and modeling efforts on soil C dynamics in regions limited by temperature and moisture.

# 1 Introduction

Carbon (C) cycle feedback to climate change is highly uncertain in current Earth System Models (ESMs) (Friedlingstein et al., 2006, Bernstein et al., 2008, Ciais et al., 2013, Bradford et al., 2016), which largely stems from their diverse simulations of C exchanges among the atmosphere, vegetation, and soil (Luo et al., 2016, Smith et al., 2016, Mishra et al., 2017). Soil organic carbon (SOC) represents the largest terrestrial carbon pool, which stores at least three times as much as the atmospheric and vegetation C reservoirs (Parry et al., 2007, Bloom et al., 2016). However, a five- to six-fold difference in soil C stocks among ESMs or offline global land surface model has been found (Todd-Brown et al., 2013, Luo et al., 2016). It is difficult to reduce or even diagnose this uncertainty, as many processes collectively affect the time of C atoms transit the soil system (i.e., transit time; $\tau_{soil}$) (Sierra et al., 2017, Spohn and Sierra, 2018,). Some recent attempts at evaluating and diagnosing the modeled SOC in ESMs have shown significant simulation uncertainties in the $\tau_{soil}$ (Todd-Brown et al., 2013, Carvalhais et al., 2014, He et al., 2016, Koven et al., 2017). For example, there is a fourfold difference in the simulated $\tau_{soil}$ among the ESMs from the $5^{th}$ phase of Coupled Model Intercomparison Project (CMIP5) (Todd-Brown et al., 2013). A recent data-driven analysis has suggested that the current ESMs have substantially underestimated the $\tau_{soil}$ by 16-17 times at the global scale (He et al., 2016). Therefore, identifying the locations of such underestimations is critical to improve the predictive ability of ESMs on terrestrial C cycle, and the construction of a benchmarking database of available observations is urgently needed (Koven et al., 2017).

The terms of transit time, turnover time and age of soil C have been muddled in diagnosing the models (Sierra et al., 2017). The diagnostic times derived from observational data are based on the different assumptions and mainly derived from four approaches. The first approach commonly defined as "*turnover time*", calculated by the division of SOC stock by C fluxes such as net primary productivity (NPP) or heterotrophic respiration ($R_h$). It assumes the soil system as a time-invariant linear system in a steady state (Bolin et al., 1973, Sanderman et al., 2003, Six and Jastrow, 2012). The second approach is based on the shifts in stable isotopic C ($^{13}C$) after successive changes in $C_3-C_4$ vegetation, together with additional information from the disturbed and undisturbed soils (Balesdent et al., 1987; Zhang et al., 2015). The third approach is based on simulating soil C dynamics with linear models by assimilating the observational data from laboratory incubations of

soil samples (Xu et al., 2016). The last approach derives the weighted inverse of the first-order cycling rate by fitting a one- or multiple-pool linear model to field observations of radiocarbon ($^{14}$C) (Trumbore et al., 1993, Fröberg et al., 2011). The diagnostic times derived from the former three approaches indicate the transit times which are the mean ages of C atoms leaving the carbon pools during the certain time (Rasmussen et al., 2016). Lu et al., (2018) has evaluated the deviation between C transit and turnover times with the CABLE model. Their results have shown that the global latitudinal pattern of C transit and turnover times are consistent under the steady-state assumption and autonomous conditions except 8% of divergence in the northern high latitudes (>60 °N). However, the diagnostic time calculated by the radiocarbon signal indicates the average age of C atoms stored in the C pools. Although radiocarbon has been widely used to quantify the age or transit time of soil C, its validity has been challenged by some recent theoretical analyses (Sierra et al., 2017, Metzler et al., 2018). Rasmussen et al., (2016) has marked off the transit time and mean system age in a mathematic way and further applied into the CASA model. Also, the methodological uncertainty is large especially when these approaches are applied to estimate the $\tau_{soil}$ of different soil fractions (Feng et al., 2016). Thus, this study mainly collects the $\tau_{soil}$ from the approaches of *stock-over-flux*, $^{13}$C changes and lab incubations in the further analyses.

In this study, we first construct a database from the literatures which reported the $\tau_{soil}$ (Fig. 1a, Supplementary materials on Text S1). Then, the database is used to evaluate the simulated $\tau_{soil}$ by the ESMs in the CMIP5. The SOC $\tau_{soil}$ were calculated under the homogenous one-pool assumption at the steady state for all studies. Data from observations and CMIP5 ensemble were then used to calculate the $\tau_{soil}$ based on both one-pool and three-pool models. Many ESMs, e.g., CESM, have released new versions in the recent years, so we also evaluate whether the simulated $\tau_{soil}$ has been improved. In the case of CESM, one of its major developments on the soil C cycling is the vertically resolved soil biogeochemical scheme (Koven et al., 2013). Thus, we employ a matrix approach developed by Huang et al., (2017) to examine the impact of the vertically resolved soil biogeochemical scheme on the simulated $\tau_{soil}$ by CESM.

**2 Materials and Methods**

**2.1 A global database of site-level $\tau_{soil}$**

We collected the literatures that reported the $\tau_{soil}$ based on measurements (Supplementary Materials on Text1): (1) $\delta^{13}C$ shifts after successive changes in $C_3-C_4$ vegetation, (2) measurements of $CO_2$ production in laboratory SOC incubation over at least seven months, and (3) simultaneously measurements of SOC stock and heterotrophic respiration (*stock-over-flux*). We constructed a database containing the measured $\tau_{soil}$ from 187 sites across the globe (Fig.1). Based on the homogenous assumption, the soil system is a time-invariant linear system at the steady state. The $\tau_{soil}$ derived from this database is under one-pool assumption. The information of climate (e.g., mean annual temperature and precipitation) was also collected from the literatures or extracted from the WorldClim database version 1.4 (http://worldclim.org/) if they were not available. The WorldClim dataset provided a set of free global climate data for ecological modelling and Geographic Information System analyzing with a spatial resolution of $0.86\ km^2$ (Hutchinson et al., 2004). We extracted the mean temperature and precipitation by averaging the monthly climate data over 1990−2000 for those observational sites with missing climate information. The classes of biomes were processed to match the seven biomes classification adopted by the MODIS land cover product MCD12C1 (NASA LP DAAC 2008, Friedl et al., 2010) and Todd-Brown et al. (2013) (Fig. S1): (1) tropical forest including evergreen broadleaf forest between 25 °N and 25 °S; (2) temperate forest including deciduous broadleaf, evergreen broadleaf outside of 25 °N and 25 °S and mixed forest south of 50 °N; (3) boreal forest including evergreen needleleaf forest, deciduous needleleaf forest, mixed forest north of 50 ° N; (4) grassland and shrubland including woody savanna south of 50 °N, savanna and grasslands south of 55 °N; (5) deserts and savanna including barren or sparsely vegetated, open shrubland south of 55 °N, and closed shrubland south of 50 °N; (6) Tundra; and (7) Croplands. Other land cover types like permanent wetland, urban, and bare land were not included in this study.

**2.2 Outputs of Earth system models from CMIP5**

The *historical* simulation outputs of 12 ESMs participating CMIP5 from 1850 to 1860 (https://esgf-data.dkrz.de/search/cmip5-dkrz/) were analyzed in this study (Table S1). For each model, the SOC, litter C, NPP, and Rh were extracted from the outputs in historical simulations (*cSoil, cLitter, npp*, and *rh,* respectively, from the CMIP5 variable list). The litter and soil carbon were summed as the bulk soil carbon stock. Among the 12 models, only the inmcm4 model did not output NPP, so we calculated it as gross primary production minus autotrophic respiration. Due to the diverse spatial resolutions among the models, we aggregated the results of different

models to $1\,°\times1\,°$ with the nearest interpolation method (Fig.S2). The $\tau_{soil}$ of SOC was calculated

as the ratio of carbon stock over flux (NPP or Rh):

$$\tau_{soil} = \frac{SOC}{flux}$$ (1)

**2.3 Estimated the SOC $\tau_{soil}$ with a three-pool model**

To examine whether the major findings of this data-model comparison is affected by the one-pool

homogenous assumption, we fitted a three-pool model with observational data and model

ensemble outputs at the biome level. In this study, a three-pool C model consisted of fast, slow,

and passive pools and carbon transfers among three pools (Fig. S3a). This model shares the same

framework with the CENTURY and the Terrestrial Ecosystem models (Bolker *et al.*, 1998; Liang

*et al.*, 2015). The dynamics of soil carbon pools follow first-order differential kinetics. The total

C stocks and $CO_2$ efflux from observations and CMIP5 ensemble were separated into pool-specific

decomposition rates by the deconvolution analysis (Fig. S3a, Liang et al., 2015). We assumed the

total soil carbon input equals to total soil respiration at the steady state.

Based on the theoretical analysis, the dynamics of the three-pool can be mathematically

described by matrix equation (Luo *et al*., 2003; Xia *et al*., 2013) as:

$$\frac{dC(t)}{dt} = I(t) - \mathbf{A}\mathbf{K}C(t)$$ (2)

where the matrix $\mathbf{C}(t) = (C_1(t), C_2(t), C_3(t))^{T}$ is used to describe soil carbon pool sizes. $\mathbf{A}$ is a matrix

given by:

$$\mathbf{A} = \begin{pmatrix} -1 & f_{12} & f_{13} \\ f_{21} & -1 & 0 \\ f_{31} & f_{32} & -1 \end{pmatrix}$$ (3)

The elements $f_{ij}$ are carbon transfer coefficients, indicating the fractions of the C entering $i$-th

(row) pool from $j$-th (column) pool. $\mathbf{K}$ is a $3\times3$ diagonal matrix indicating the decomposition rates

(the amounts of C per unit mass leaving each of the pools per year). The matrix of $\mathbf{K}$ is given by:

$\mathbf{K} = \text{diag}\,(k_1, k_2, k_3)$.

The parameters in the three-pool model were estimated based on Bayesian probabilistic

inversion (equation (4)). The posterior probability density function $P(\theta|Z)$ of model parameters

($\theta$) can be represented by the prior probability density function $(P(\theta))$ and a likelihood function

(P(Zlθ)) (Liang *et al.*, 2015; Xu *et al.*, 2016). The likelihood function was calculated by the

minimum error between observed and modelled values with equation (5). In this study, we adopted

the prior ranges of model parameter from Liang et al. (2015).

$P(\theta|Z) \propto P(Z|\theta) \cdot P(\theta)$                                              (4)

$P(Z|\theta) \propto \exp\left\{-\frac{1}{2\sigma_i^2(t)} \sum_{i=1}^{n} \sum_{t \in \text{obs}(Z)} [Z_i(t) - X_i(t)]^2\right\}$                 (5)

where $Z_i(t)$ and $X_i(t)$ are the observed and modelled transit times, and the $\sigma_i^2(t)$ is the standard

deviation of measurements. The posterior probability density function of the parameters was

constructed with two steps: a proposing step and a moving step. In the first step, the dataset was

generated based on the previously accepted data with a proposal distribution:

$\theta^{\text{new}} = \theta^{\text{new}} + \frac{d(\theta_{\max} - \theta_{\min})}{D}$                                         (6)

where $\theta_{\max}$ and $\theta_{\min}$ are the maximum and minimum values of the given parameters, $d$ is the

random variable between -0.5 and 0.5 with uniform distribution, $D$ is used to control the proposing

step size in this study. In the moving step, the new data $\theta^{\text{new}}$ is tested against the Metropolis criteria

to quantify whether it should be accepted or rejected. The parameters of posterior probability

density function were constructed by the Metropolis-Hasting algorithm. The Metropolis-Hasting

algorithm was run 50,000 times for observed data. Accepted parameter values were used in the

further analysis.

Based on the concepts of mean age and mean transit time published by Rasmussen et al., (2016)

and Lu et al., (2018), the mean carbon age defined as the whole time periods the carbon atoms

stored in the carbon pools, and then the mean age of carbon $\bar{a}_i(t)$ in a certain carbon pool $i$ could

be calculated with equation (7):

$\bar{a}_i(t) = 1 + \frac{\sum_{i=1}^{3} (\bar{a}_j(t) - \bar{a}_i(t)) \cdot f_{ij}(t) \cdot C_i - \bar{a}_j(t) \cdot I_i(t)}{C_i}$                       (7)

where the $f_{ij}(t)$ are the carbon fraction transfer coefficients from $j$-th to $i$-th pools, $I_i(t)$ is the

external input into the $i$-th carbon pool. The transit time $\tau_i(t)$ was defined as the mean age of

carbon atoms leaving the carbon pool at a specific time:

$\tau_i(t) = \sum_{i=1}^{d} f_i(t) \cdot a_i(t)$                                                    (8)

where the $f_i(t)$ is the fraction of carbon with mean age $a_i(t)$.

**2.4 Matrix approach through CLM4.5 and CLM4.5_noV**

The Community Land Model Version 4.5 (CLM4.5) is the terrestrial component of Community

Earth System Model (CESM). This version mainly consists of exchanges among different carbon

and nitrogen pools and other biogeochemical cycles, as well as includes a vertical dimension of

soil carbon and nitrogen transformations (Koven et al., 2013). The matrix approach was applied to

extract the soil module from original CLM4.5 which could evaluate which processes influence $\tau_{soil}$

in the model (Huang et al., 2017). Once get the total C pool and $R_h$ in each pool, we can calculate

the $\tau_{soil}$ with the equation (1). We represented the structure of SOC as 7 carbon pools as *i)* one

coarse woody debris (CWD) pool, *ii)* three litter pools (litter1, litter2 and litter3) and *iii)* three soil

carbon pools (soil1, soil2, and soil3). In this matrix, C is transferred from three litter pools and

CWD to three soil pools with different transfer rates. In each layer, these transfer rates are regulated

by the transfer coefficients and fractions. C inputs from litterfall were allocated into different C

compartments by modifications by soil environmental factors (temperature, moisture, nitrogen and

soil oxygen) and vertical transfer process. To understand whether the incorporation of soil vertical

profile affect the simulation of $\tau_{soil}$, we compared the results based on matrix approach with (i.e.,

CLM4.5) or without (i.e., CLM4.5_noV) the soil vertical transfer process.

In the CLM4.5, soil C dynamics was simulated with 10 soil layers, and the same organic

matter pools among different vertical soil layers are allowed to mix mainly through diffusion and

advection. The matrix approach determinates the soil dynamic of each SOC pool by simulating

the first-order kinetics as equation (9):

$$\frac{d\mathbf{C}(t)}{dt}=\mathbf{B}(t)I(t)-\mathbf{A}\xi(t)\mathbf{K}\mathbf{C}(t)-\mathbf{V}(t)\mathbf{C}(t) \tag{9}$$

where the $\mathbf{C}_{(t)}$ is the organic C pool size at time *t*. $I_{(t)}$ is the total organic C inputs while $\mathbf{B}_{(t)}$ is the

vector of partitioning coefficients. $\mathbf{K}$ is a diagonal matrix which representing the intrinsic

decomposition rate of each C pool. The decomposition rate in the matrix approach is modified by

the transfer matrix $\mathbf{A}$ and environmental scalars $\xi$. The scalar matrix $\xi$ shown in equation (10) is

the environmental factor to modify the SOC intrinsic decomposition rate. Each scalar matrix

combines temperature ($\xi_T$), water ($\xi_W$), oxygen ($\xi_O$), depth ($\xi_D$) and nitrogen ($\xi_N$) controlled scalar

on SOC decay.

$\xi'=\xi_T\xi_W\xi_O\xi_D\xi_N$                                            (10)

**A** is the horizontal C transfer matrix which quantifies C movement among different C pools shown

as matrix (10). The non-diagonal entries $A_{ij}$ shown in matrix (10) represent the fraction of C

moves from the $j$-th to the $i$-th pool. In CLM4.5 and CLM4.5_noV, transfer coefficients are the

same in each soil layer.

$$\mathbf{A} = \begin{pmatrix} 0 & 0 & 0 & 0 & 0 & 0 & 0 \\ 0 & 0 & 0 & 0 & 0 & 0 & 0 \\ 0 & 0 & 0 & 0 & 0 & 0 & 0 \\ 0 & 0 & 0 & f_{44} & 0 & 0 & 0 \\ 0 & f_{52} & f_{53} & 0 & f_{55} & f_{56} & f_{57} \\ 0 & 0 & 0 & f_{64} & f_{65} & f_{66} & 0 \\ 0 & 0 & 0 & 0 & f_{75} & f_{76} & f_{77} \end{pmatrix}$$      (11)

$\mathbf{V}(t)$ is the vertical C transfer coefficient matrix among different soil layers, each of the diagonal

blocks is a tridiagonal matrix that describes transfers coefficient with $\mathbf{V}_{ij}(t)$. In this section,

CLM4.5_noV assumes no vertical transfers in all pools. Therefore, $\mathbf{V}(t)$ for CLM4.5_noV is a

blank matrix in the simulation. In the contrast, CLM4.5 was assigned by a matrix with vertical

transfers in each C pool. As the vertical transfer rates among different C pool categories in CLM4.5,

the matrix shown as matrix (12).

$$\mathbf{V}(t) = \begin{pmatrix} 0 & 0 & 0 & 0 & 0 & 0 & 0 \\ 0 & V_{22}(t) & 0 & 0 & 0 & 0 & 0 \\ 0 & 0 & V_{33}(t) & 0 & 0 & 0 & 0 \\ 0 & 0 & 0 & V_{44}(t) & 0 & 0 & 0 \\ 0 & 0 & 0 & 0 & V_{55}(t) & 0 & 0 \\ 0 & 0 & 0 & 0 & 0 & V_{66}(t) & 0 \\ 0 & 0 & 0 & 0 & 0 & 0 & V_{77}(t) \end{pmatrix}$$      (12)

**2.4 Statistical analyses.**

The median and interquartile were used for the quantification of both observational and modelling

results due to the probability distribution of $\tau_{soil}$ is not normal. To test the difference in $\tau_{soil}$ among

three approaches, we first normalized the data with the log-transformation and then applied the one-way ANOVA with multi-comparison technique (Fig. 1b insert). The linear regression and correlation analyses were performed in *R* (3.2.1; *R* development Core team, 2015).

The Gaussian kernel density estimation was used to obtain the distributions of observed transit times (Sheather & Marron, 1990; Saoudi *et al.*, 1997). The Gaussian kernel density estimation is a non-parametric approach to estimate the probability density function of a random variable. Let $(x_1, x_2, \cdots, x_n)$ denote the observed SOC $\tau_{soil}$ with density function $f$ as below:

$$\hat{f}_h(x) = \frac{1}{nh}\sum_{i=1}^{n} K(\frac{x-x_i}{h}) \tag{13}$$

where $K$ is the non-negative function than integrates to one and has mean zero, and $h > 0$ is a smoothing parameter called the bandwidth. The bandwidth for approaches of stable isotope [13]C, *stock-over-flux* and incubation are: 48.61, 35.13, 2.62, respectively.

**3 Results and discussion**

**3.1 $\tau_{soil}$ and its spatial variation by different approaches**

The one-way ANOVA with multi-comparison analysis showed no significant difference in the log-transformed $\tau_{soil}$ between the methods of [13]C (Median = 60 years; interquartile range = 8 to 29 years) and *stock-over-flux* (16; 3 to 156 years, Fig. 1b). The range of these field *in-situ* measurements (31; 5 to 84 years) is comparable to a former estimate of mean SOC turnover time (48 with 24 to 107 years) across twenty long-term experiments in temperate ecosystems using the [13]C labelling approach (Schmidt et al., 2011). However, the estimates of $\tau_{soil}$ from laboratory studies (4; 1 to 15 year) was significantly shorter than the other two methods (Fig. 1b). It suggests that the $\tau_{soil}$ could be underestimated by the measurements from the laboratory incubations studies. Thus, the $\tau_{soil}$ from the laboratory incubation studies were excluded in the following analyses.

We then integrated the estimates of $\tau_{soil}$ based on the [13]C, and *stock-over-flux* approaches to examine the inter-biome difference. As shown by Figure 2b, the longest $\tau_{soil}$ was found in desert and shrubland (170; 58 to 508) and tundra (159; 39 to 649 years). Boreal forest (58; 25 to 170 years) has longer $\tau_{soil}$ than the temperate (44; 13 to 89 years) and tropical forests (15; 9 to 130 years). Grassland and savanna had short (35; 21 to 57 years) and croplands had moderate (62; 21 to 120 years) $\tau_{soil}$ in comparison with other biomes (Fig. 2).

**3.2 Modelled $\tau_{\text{soil}}$ in the CMIP5 ensemble and its estimation biases**

The longest ensemble mean $\tau_{\text{soil}}$ of multiple models were found in dry and cold regions (Fig. 2). In comparison with the integrated observations from [13]C and stock over flux, the modelled $\tau_{\text{soil}}$ were significantly shorter across all biomes (Fig. 2b insert). The negative bias was larger in dry (desert, grassland, and savanna) and cold (tundra and boreal forest) regions than tropical and temperate forests. The longest modelled $\tau_{\text{soil}}$ appeared in the tundra ecosystem with the median of 64 years. The modelled median $\tau_{\text{soil}}$ were also shorter than observations in tropical forest (9 years), temperate forests (13 years), boreal forest (24 years), grassland/savanna (25 years), desert and shrubland (58 years) and croplands (27 years) (Fig. 2). In comparison with the observations, the models obviously underestimated the $\tau_{\text{soil}}$ in the cold and dry biomes (Fig. 2b). A recent global data-model comparison study at the 0.5 $°\times$0.5 $°$resolution has also detected a similar spatial pattern of underestimation bias in ecosystem C turnover time (Carvalhais et al., 2014), but its magnitudes of bias in the cold regions are much smaller than that found in this study.

By grouping the $\tau_{\text{soil}}$ into different climatic categories, we found that the observed $\tau_{\text{soil}}$ significantly covaried with MAT ($y = -5.28x+156.04$, $r^2 = 0.48$, $P <0.01$) and MAP ($y= -68.19x+1222.6$, $r^2 = 0.60$, $P<0.01$) (Fig. 3). These results support the previous findings of negative covariations between $\tau_{\text{soil}}$ and temperature at both the site and global levels (Trumbore *et al.* 1996). Although there is no significant correlation between $\tau_{\text{soil}}$ and MAP in the observations, the models produced negative correlations of $\tau_{\text{soil}}$ with MAT ($r^2 = 0.24$, $P < 0.05$) and MAP ($r^2 = 0.44$, $P < 0.05$) (Fig. 3).

**3.3 Estimation the $\tau_{\text{soil}}$ with a three-pool model**

With the three-pool model, the total C stocks and $CO_2$ efflux from observations and CMIP5 ensemble were separated into pool-specific decomposition rates by the deconvolution analysis (Fig. S3a, Liang et al., 2015). Seven out of eleven parameters were constrained for tropical forest and cropland (Fig. S4, Fig. S9). Eight out of eleven parameters were constrained for temperate, boreal forest and desert & shrubland (Fig. S5, S6, S8). Five out of eleven parameters were constrained for tundra ecosystem (Fig. S7). For grassland and savanna, seven out of eleven parameters were constrained (Fig. S10).

The longest simulated $\tau_{soil}$ appeared in tundra (167 years) and desert (135 years) (Fig. 4, Table S3). Temperate forest (79 years) has longer $\tau_{soil}$ than the boreal (66 years) and tropical forests (29 years). Grassland and savanna had short (53.8 years) and croplands had moderate (77 years) $\tau_{soil}$ in comparison with other biomes. The $\tau_{soil}$ calculated from the one- and three-pool models did not show large difference across all biomes. Also, estimates based on these two model structures showed the largest underestimation of $\tau_{soil}$ in the tundra and desert (Fig. 4).

**3.4 Improved modeling of $\tau_{soil}$ with vertically resolved SOC dynamics**

Given that many ESMs have further developed their representations of the soil biogeochemistry in recent years, we also examined whether the $\tau_{soil}$ estimates have been improved by one of the CMIP5 models (i.e., CESM). It is encouraging that the biases of $\tau_{soil}$ in dry and cold regions have been substantially reduced in the new land version of CESM (i.e., version 4.5 of the Community Land Model; CLM4.5). One major improvement in CLM4.5 is the vertically resolved SOC dynamics (Koven et al., 2013). The soil organic carbon is allowed to transfer through diffusion and advection up to 3.8 m within 10 layers. In each layer, the transfer rates are regulated by the environmental scalars (i.e. temperature, soil moisture and available oxygen). The $\tau_{soil}$ simulated by CLM4.5 are longer than CLM4 (with median value 137 year & 21 year) especially in northern high latitudinal regions. By turning off the vertical C movements with a matrix approach (i.e., there is no vertical C transfer, thus, the vertical matrix is a zero matrix in equation (12)), we showed a similar pattern of underestimation on $\tau_{soil}$ by CLM4.5 (i.e., CLM4.5_noV in Fig. 5). Huang et al., (2017) also reported the longer $\tau_{soil}$ and high carbon storage capacity in northern high latitudes. Those result suggest that the vertically resolved soil biogeochemistry is promising in improving the $\tau_{soil}$ estimates by ESMs. However, it should be noted that the spatial variation of $\tau_{soil}$ is still largely underestimated by the CLM4.5 (Fig. 5b insert).

Higher NPP values simulated by ESMs in the cold and dry regions have been reported by previous studies (Shao et al., 2013, Smith et al., 2016, Xia et al., 2017). The models produce high NPP in cold regions largely because they overestimate the efficiency of plant transferring assimilated C to growth (Xia et al., 2017). The CMIP5 models overestimate the precipitation and underestimate the dryland expansion by 4 folds during 1996-2005 (Ji et al., 2015), which could lead to high NPP and fast SOC turnover rates. These results suggest that once the NPP simulation

is improved without the correction of the $\tau_{soil}$ underestimation, the models will produce smaller SOC stock in the cold and dry ecosystems.

This study shows that adding the vertical resolved biogeochemistry is a promising approach to correct the bias of $\tau_{soil}$ in current models. However, other processes such as the microbial dynamics, SOC stabilization and nutrient cycles could affect the estimation of $\tau_{soil}$, but are so far fully considered by the CMIP5 models (Luo et al., 2016). For example, adding soil microbial dynamics could increase $\tau_{soil}$ in cold regions by lowering the transfer proportion of decomposed SOC to the atmosphere (Wieder et al., 2013). By contrast, the incorporation of nitrogen cycles might shorten $\tau_{soil}$ by increasing plant C transfers to short-lived litter pools (e.g., O-CN and CABLE model) (Gerber et al., 2010) or reducing litter C transfers to the slow soil C pools (e.g., LM3V model) (Xia et al., 2013).

Large challenges still exist in using observations derived from different methods to constrain the modelled $\tau_{soil}$. Laboratory incubation studies report much shorter $\tau_{soil}$ than other methods, mainly due to the optimized soil moisture and/or temperature during the soil incubation (Stewart et al., 2008; Feng et al., 2016). It suggests that the ESM models will largely underestimate $\tau_{soil}$ if its turnover parameters are derived from laboratory incubation studies. It should be noted that the observations from the [13]C and the *stock-over-flux* approaches in this study are derived for the bulk soil. However, SOC is commonly represented as multiple pools with different cycling rates in most of the CMIP5 models (Luo et al., 2016, Sierra et al., 2017, 2018, Metzler and Sierra, 2018). As synthesized by Sierra et al. (2017), the observations of $\tau_{soil}$ are useful for a specific model once its pool structure is identified. This study also detect difference in the estimated $\tau_{soil}$ between the one- and three-pool models (Fig. 4). Thus, model database, such as the bgc-md (https://github.com/MPIBGC-TEE/bgc-md), is a useful tool to improve the integration of observations and soil C models. An enhanced transparency of C-cycle model structure in ESMs is highly recommended, especially when they participate in the future model intercomparison projects such as the CMIP6 (Jones et al., 2016).

**4 Conclusions**

This study detected large underestimation biases of $\tau_{soil}$ in ESMs in cold and dry biomes, especially the tundra and desert. Improving the modelling of SOC dynamics in these regions is important because the cold ecosystems (e.g., the permafrost regions) are critical for global C feedback to

future climate change (Schuur et al., 2015) and the dry regions strongly regulate the interannual variability of land $CO_2$ sink (Poulter et al., 2014, Ahlström et al., 2015). The current generation of ESMs represents the soil C processes with a similar model formulation as first-order C transfers among multiple pools (Sierra et al., 2015, Luo et al., 2016, Metzler and Sierra, 2018). Thus, tremendous research efforts are still required to attribute the underestimation biases of $\tau_{soil}$ in current ESMs to their sources, such as the model structure, parameterization, and climate forcing. Reducing these biases would largely improve the accuracy of ESMs in the projection of future terrestrial C cycle and its feedback to climate change. Recent modelling activities aiming to increase the soil heterogeneity, e.g., soil vertical profile (Koven et al., 2013, 2017) and microbial dynamics (Allison et al., 2010, Wieder et al., 2013), are promising. Overall, this study shows the great spatial variation of $\tau_{soil}$ in the natural ecosystems, and we recommend more research efforts to improve its representation by ESMs in the future.

## 5 Acknowledgments

We appreciated the anonymous reviewers for their valuable suggestions. We also appreciated Dr. Todd-Brown for her supports of the soil data in CMIP5, and Dr. Deli Zhai for the valuable comments. The model simulations analyzed in this study were obtained from the Earth System Grid Federation CMIP5 online portal hosted by the Program for Climate Model Diagnosis and Intercomparison at Lawrence Livermore National Laboratory (https://pcmdi.llnl.gov/projects/esgf-llnl/). This work was financially supported by the National Natural Science Foundation (31722009, 31800400, 41630528), the National Key R&D Program of China (2017YFA0604603), the Fok Ying-Tong Education Foundation for Young Teachers in the Higher Education Institutions of China (Grant No. 161016), and the National 1000 Young Talents Program of China.

## 6 Author information and contributions

The authors declare no competing financial interests. Correspondence should be addressed to J. Xia (jyxia@des.ecnu.edu.cn). J.X designed the study. J.W collected and organized the data. L. J provided the CMIP5 and HWSD data. X. X provided the laboratory incubation data. Y. Huang provides the CLM4.5 matrix module. J.W and J.X wrote the first draft, and all other authors contributed to the revision and discussions on the results.

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

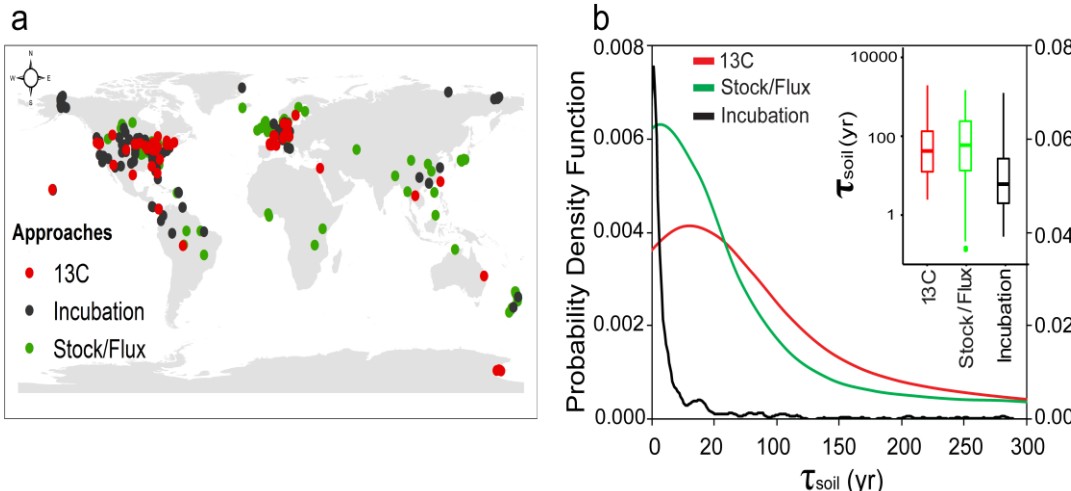

Figure 1. Spatial distributions of observational sites for estimates of SOC transit time ($\tau_{soil}$, year). (**a**), The site locations of measurements with different approaches. (**b**), Probability density functions of $\tau_{soil}$ measured by different approaches. Note that the left axis is for $^{13}$C and *stock-over-flux* approaches, and the right axis is for laboratory incubation studies.

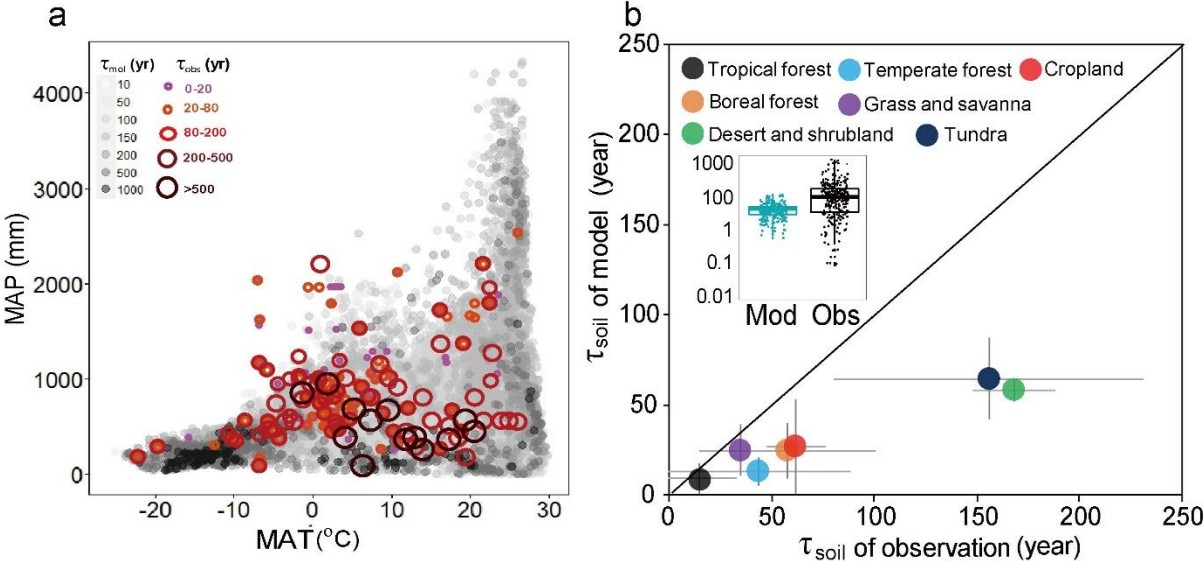

Figure 2. Global spatial variation of SOC transit time ($\tau_{soil}$) with climate and the difference of $\tau_{soil}$ estimation between observations and models. (**a**), Spatial variation of $\tau_{soil}$ with mean annual temperature (MAT) and mean annual precipitation (MAP). (**b**), Comparisons of modelled against observed $\tau_{soil}$. Details for the classification of biomes are provided in the method section.

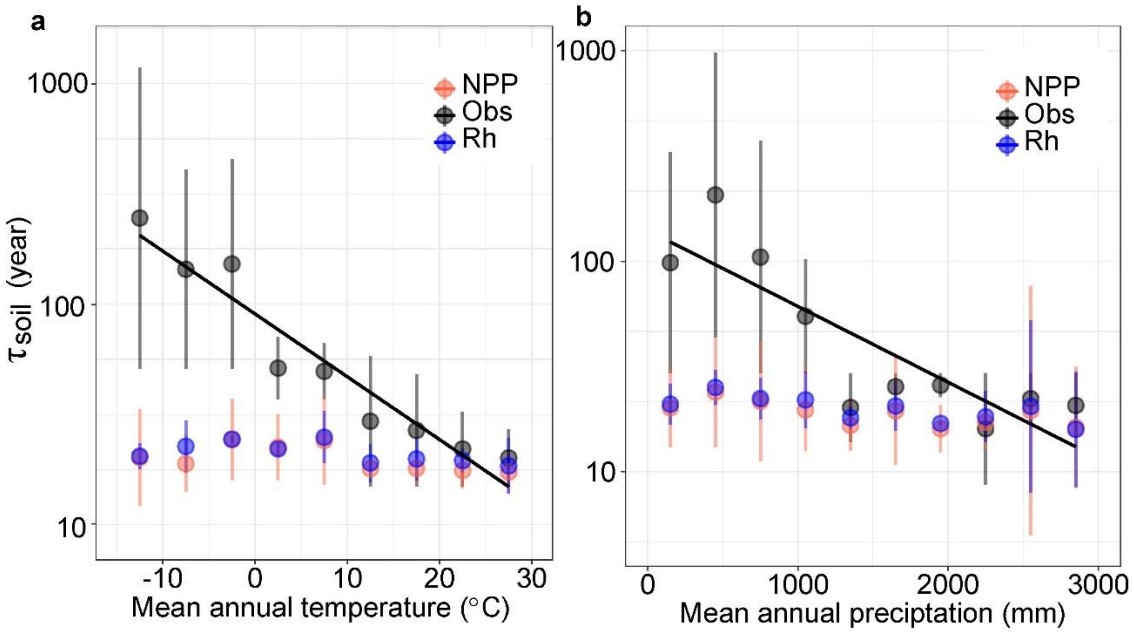

Figure 3. Relationships between SOC transit time ($\tau_{soil}$) and climate factors in both observations and CIMP5 models. The black solid lines show the negative correlation between $\tau_{soil}$ and (**a**) mean annual temperature and (**b**) mean annual precipitation. The black dots indicate the aggregated $\tau_{soil}$ over each category of MAT ($y= -5.47x+1971.5$, $r^2 = 0.49$, $P<0.01$) or MAP ($y= -68.19x+1222.6$, $r^2 = 0.60$, $P<0.01$). The red and blue dots present the mean value of the multiple models based on the ratios of carbon stock over NPP and $R_h$, respectively.

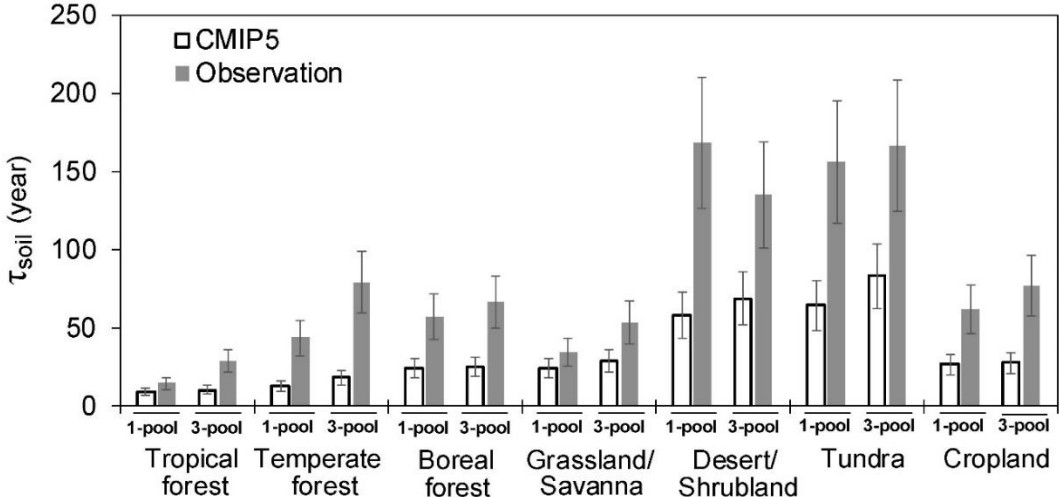

Figure 4. The SOC transit time ($\tau_{soil}$) calculated from the one- and three-pool models under the steady-state assumption.

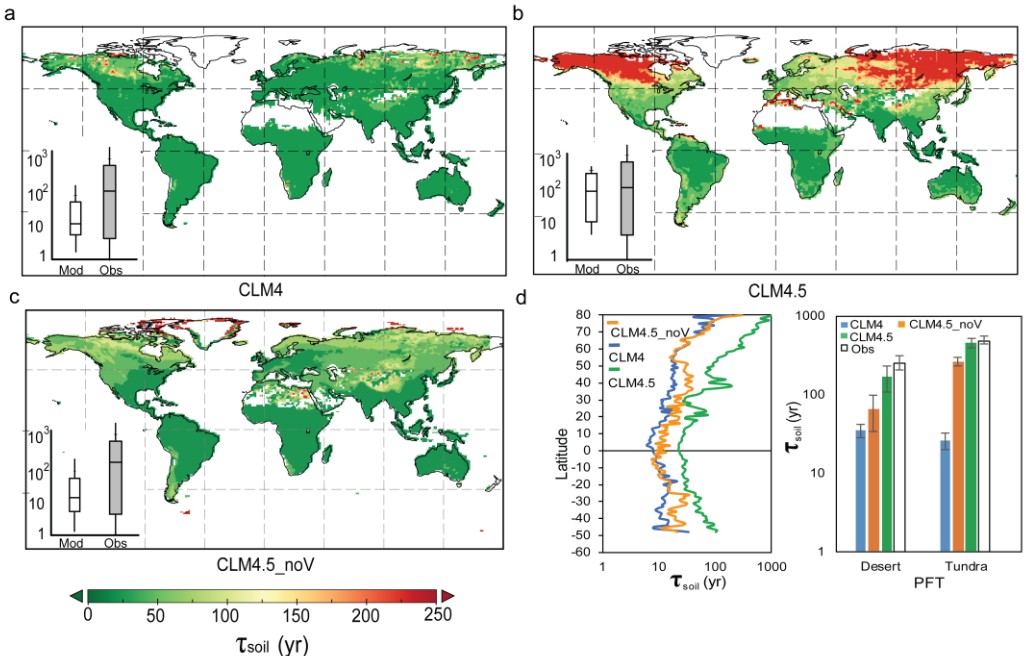

Figure 5. Simulated SOC transit time ($\tau_{soil}$) by CLM4 (**a**; median global $\tau_{soil}$ = 20.56 years), CLM4.5 (**b**; median global $\tau_{soil}$ =127.50 years) and CLM4.5_noV (**c**; median global $\tau_{soil}$ =22.24 years). The panel (**d**) shows the latitudinal spatial distribution of the mean $\tau_{soil}$ of different models in desert and tundra. The insert figures in panels a-c compare the $\tau_{soil}$ between models and observations. The bottom and top of the box represent the first and third quartiles.

