# Peer review of "Evaluating the simulated mean soil carbon transit times by Earth System Models using observations"

_Biogeosciences, 2018_

## Referee Comment (RC1) · Anonymous Referee #1 · 11 Sep 2018

In this paper, the authors evaluate soil carbon transit times in 12 CMIP5 models. They found that, compared to in-situ observations, transit times are usually underestimated by models, especially in cold regions and dry/hot regions. The authors show that some of these biases can be resolved by adopting more vertically-resolved parameterization of soil C dynamics with the CLM4.5 model.

I have concerns about this manuscript as it seems very similar to previous papers by e.g. Todd-Brown et al. (2013): the same models are evaluated with the same HWSD-MODIS based product. The novelty here is the comparison of models against transit times measured in worldwide soils, and I think it should be the main aim of the study. If the authors decide to keep the global evaluation, the HWSD-MODIS product should be confronted to in situ observations to justify its use as a global benchmark or,

alternatively, the creation of this database could be used to derive a more robust global product.

Section 3.2 is very hard to understand. It is not clear whether models are evaluated against the in-situ observations, or whether they are evaluated against the HWSD-MODIS based global product (as it seems in Figure 3). The discussion around improvements due to the addition of a vertical resolution in CLM4.5 is reduced to less than 10 line while it seems to be one of the key findings of the whole study.

Hereafter are some more detailed comments:

p3 l 21-29: which period of the historical simulation did the authors consider?

p3 l30: I find that there is a missed opportunity here to use in situ observations to derive a more robust global dataset of transit times. HWSD and MODIS NPP both come with known biases and there may be other products to choose from e.g. soilgrids (www.soilgrids.org)

p6 l30-35: I do not understand what is learned by replacing MODIS NPP with TRENDY models (which ones? reference is missing here). Does that mean that TRENDY is considered as an observation of NPP against which ESMs are evaluated?

Figure 2: from the legend, panels c and d are missing. Panel a is hard to understand and uncertainties are missing from panel b.

Figure 3: in panel a and b, do black dots represent data from the 187 sites? or were they extracted from the HWSD/MODIS product?

references

Todd-Brown, K.E., Randerson, J.T., Post, W.M., Hoffman, F.M., Tarnocai, C., Schuur, E.A. and Allison, S.D.: Causes of variation in soil carbon simulations from CMIP5 Earth system models and comparison with observations. Biogeosciences, 10, 1717-1736, https://doi: 10.5194/bg-10-1717-2013.

---

## Referee Comment (RC2) · Anonymous Referee #2 · 15 Oct 2018

In this manuscript, the authors present observation-based estimates of transit times of carbon in soils, and compare these estimates with model predictions. This is an important topic because transit times are a very good constraint for evaluating model performance. There has been a lot of recent research on this topic, motivated in part by the work of Carvalhais et al. (2014), who used an *stock-over-flux* approach to compute residence times from models and observations. Recent publications have shown that this approach has problems to compute transit times for systems of multiple pools and out of equilibrium (Lu et al., 2018; Sierra et al., 2017), and better methods for estimating transit times for systems out of equilibrium have been developed (e.g. Rasmussen et al., 2016).

Despite these recent developments, this manuscript uses observations from incubation

experiments and $\delta^{13}$C measurements from C$_3$/C$_4$ vegetation replacement experiments, in which the rate of soil carbon loss is estimated assuming one single pool in equilibrium. This is evidenced by equations (1) to (3) in Text S1 of the supplementary material. The implication of this assumption is that the observations are treated as a homogeneous system, without differentiating between the age of the stored carbon and the age of the carbon in the output flux. In the introductory paragraphs, the manuscript gives the impression that it provides an advance by providing observation estimates of transit times, but in reality these estimates suffer the same problems of previous approaches.

I recommend the authors to use the data they compiled to fit multiple-pool models to better estimate age and transit times from these observations. You probably would still need to keep the steady-state assumption for this type of observations, but at least you can remove the one-single homogeneous pool assumption. For a fair comparison with the model output, I recommend you compute their transit time at the spin-up state, which better represent the equilibrium state of the model. In the current version, you compute model-derived transit times from a multi-year average, but this corresponds to a transient state where transit times are not unique.

Another aspect that requires clarification is the computation of the transit time distributions in Figure 1b. How were these distributions obtained from the data? Did you assume a specific distribution function and fitted its parameter values using the data? This seems to be the case for the $\delta^{13}$ and the stock/flux data, but not for the incubations. Please clarify.

**References**

Carvalhais, N., Forkel, M., Khomik, M., Bellarby, J., Jung, M., Migliavacca, M., Mu, M., Saatchi, S., Santoro, M., Thurner, M., et al. (2014). Global covariation of carbon turnover times with climate in terrestrial ecosystems. *Nature*, 514(7521):213.
Lu, X., Wang, Y.-P., Luo, Y., and Jiang, L. (2018). Ecosystem carbon transit versus turnover

times in response to climate warming and rising atmospheric $CO_2$ concentration. *Biogeosciences Discussions*, 2018:1–22.

Rasmussen, M., Hastings, A., Smith, M. J., Agusto, F. B., Chen-Charpentier, B. M., Hoffman, F. M., Jiang, J., Todd-Brown, K. E. O., Wang, Y., Wang, Y.-P., and Luo, Y. (2016). Transit times and mean ages for nonautonomous and autonomous compartmental systems. *Journal of Mathematical Biology*, 73(6):1379–1398.

Sierra, C. A., Müller, M., Metzler, H., Manzoni, S., and Trumbore, S. E. (2017). The muddle of ages, turnover, transit, and residence times in the carbon cycle. *Global Change Biology*, 23(5):1763–1773.

---

## Author Comment (AC1) · 11 Nov 2018

The authors thank all reviewers for the useful feedback on this manuscript. Some of those suggestion could improve my manuscript. We responded to the comments in blue below, and we hope we could address the concerns from reviewers.

Reviewer 1: In this paper, the authors evaluate soil carbon transit times in 12 CMIP5 models. They found that, compared to in-situ observations, transit times are usually underestimated by models, especially in cold regions and dry/hot regions. The authors show that some of these biases can be resolved by adopting more vertically-resolved parameterization of soil C dynamics with the CLM4.5 model.

Response: Thanks for the clear summary of our manuscript.

[Figure]

Major remarks:

1) I have concerns about this manuscript as it seems very similar to previous papers by e.g. Todd-Brown et al. (2013): the same models are evaluated with the same HWSD-MODIS based product. The novelty here is the comparison of models against transit times measured in worldwide soils, and I think it should be the main aim of the study. If the authors decide to keep the global evaluation, the HWSD-MODIS product should be confronted to in situ observations to justify its use as a global benchmark or, alternatively, the creation of this database could be used to derive a more robust global product. Response: We agree that Todd-Brown et al., (2013) has done the wonderful evaluation on the large uncertainty of soil C turnover time based on the HWSD-MODIS products and 12 CMIPS models. As pointed out by the reviewer, the unique contribution of our study is using the in-situ observations to benchmark the global models. In order to avoid the confusion, we will follow the reviewer's suggestion to remove the results based on HWSD-MODIS products (i.e., panels c and d in the Fig. 3) in the revised version. We also fitted a three-pool model with the observations in the revised version. Please see the updated Figure 3 as below:

Figure R1. Relationships between transit time (log) and climate factors in both observations and CIMP5 models. The black solid lines show the negative correlation between $\tau$soil and (a) mean annual temperature and (b) mean annual precipitation. The black dots indicate the aggregated $\tau$soil over each category of MAT (y= -5.47x+1971.5, r2 = 0.49, P<0.01) or MAP (y= -68.19x+1222.6, r2 = 0.60, P<0.01). The red and blue dots present the mean value of the multiple models based on the ratios of carbon stock over NPP and Rh, respectively.

2) Section 3.2 is very hard to understand. It is not clear whether models are evaluated against the in-situ observations, or whether they are evaluated against the HWSD-MODIS based global product (as it seems in Figure 3). The discussion around improvements due to the addition of a vertical resolution in CLM4.5 is reduced to less than 10 lines while it seems to be one of the key findings of the whole study. Response:

[Figure]

The Section 3.2 was mainly the evaluation of models against the in-situ observations. In this version, we will make this section clearer by: (1) We will add more details about the comparison between model results and the in-situ observations. In brief, only the grids containing the locations of in-situ observations were selected from the models for the comparison. (2) To avoid confusion, we will remove the HWSD/MODIS results in this version. (3) The original results will be replaced with the new results based on the 3-pool model. (4) The results based on the vertical resolution in CLM4.5 will be expanded.

Specific Comments:

p3 l 21-29: which period of the historical simulation did the authors consider? Response: The historical period is from 1995 to 2005. We have made it clear in the revision.

p3 l30: I find that there is a missed opportunity here to use in situ observations to derive a more robust global dataset of transit times. HWSD and MODIS NPP both come with known biases and there may be other products to choose from e.g. soilgrids (www.soilgrids.org). Response: As mentioned above, we will remove the results based on HWSD and MODIS in the revised version. We thank the reviewer for the suggestion of deriving a robust global dataset of transit time based on the observations. This task is scientifically very important, but is difficult at the current stage due to a few reasons. First, the available observations is limited by the unequal quality and the uneven spatial distribution of the locations. Second, no data-driven approach is ready for deriving a global dataset of C transit time based on the observations. Third, it is difficult to reduce the methodological uncertainty of data (e.g., Fig. 1b) in integrating them into a given model for global calculation. We will discuss this issue in the revised manuscript.

p6 l30-35: I do not understand what is learned by replacing MODIS NPP with TRENDY models (which ones? reference is missing here). Does that mean that TRENDY is considered as an observation of NPP against which ESMs are evaluated? Response:

The results from TRENDY and MODIS NPP will be removed in this version. Also, we agree with the reviewer that TRENDY NPP cannot used as observations.

Figure 2: from the legend, panels c and d are missing. Panel a is hard to understand and uncertainties are missing from panel b. Response: Sorry for the confusion. We will correct the figure legend in the revised version. More sentences will be added to explain the panel a, and the uncertainties will be added in the panel b.

Figure 3: in panel a and b, do black dots represent data from the 187 sites? or were they extracted from the HWSD/MODIS product? Response: The black dots represent data from 187 sites in panel a and b in Figure 3, we grouped them into different levels of climatic variables. We will revise the figure legend to make it clearer. Also, the panels c and d will be removed to avoid confusion.

References Todd-Brown, K.E., Randerson, J.T., Post, W.M., Hoffman, F.M., Tarnocai, C., Schuur, E.A. and Allison, S.D.: Causes of variation in soil carbon simulations from CMIP5 Earth system models and comparison with observations. Biogeosciences, 10, 1717-1736, https://doi: 10.5194/bg-10-1717-2013.

Please also note the supplement to this comment:
https://www.biogeosciences-discuss.net/bg-2018-342/bg-2018-342-AC1-supplement.pdf

—————————————————

a

[Figure]

**Fig. 1.** Relationships between transit time (log) and climate factors in both observations and CIMP5 models.

---

## Author Comment (AC2) · 11 Nov 2018

Dear editor:

The authors thank all reviewers for the useful feedback on this manuscript. Some of those suggestion could improve my manuscript. We responded to the comments below, and we hope we could address the concerns from reviewers.

In this manuscript, the authors present observation-based estimates of transit times of carbon in soils and compare these estimates with model predictions. This is an important topic because transit times are a very good constraint for evaluating model performance. There has been a lot of recent research on this topic, motivated in part by the work of Carvalhais et al. (2014), who used a stock-over-flux approach to compute

residence times from models and observations. Recent publications have shown that this approach has problems to compute transit times for systems of multiple pools and out of equilibrium (Lu et al., 2018; Sierra et al., 2017), and better methods for estimating transit times for systems out of equilibrium have been developed (e.g. Rasmussen et al., 2016).

Response: We thank the reviewer for the great summary. We will revise the introduction of our manuscript to highlight these milestone works as mentioned by the reviewer.

Major remark:

1) Despite these recent developments, this manuscript uses observations from incubation experiments and 13C measurements from C3/C4 vegetation replacement experiments, in which the rate of soil carbon loss is estimated assuming one single pool in equilibrium. This is evidenced by equations (1) to (3) in Text S1 of the supplementary material. The implication of this assumption is that the observations are treated as a homogeneous system, without differentiating between the age of the stored carbon and the age of the carbon in the output flux. In the introductory paragraphs, the manuscript gives the impression that it provides an advance by providing observation estimates of transit times, but in reality, these estimates suffer the same problems of previous approaches. I recommend the authors to use the data they compiled to fit multiple-pool models to better estimate age and transit times from these observations. You probably would still need to keep the steady-state assumption for this type of observations, but at least you can remove the one-single homogeneous pool assumption.

Response: Many thanks for the thoughtful suggestion. We followed the reviewer's suggestion to fit the data to a three-pool model instead of the single pool approach. Then, we estimated the C transit time and age from the observations under the steady-state assumption. The estimated parameters and new results could be found as below in the Table R1 and R2, respectively. Please also find the details of the 3-pool model in supplementary material.

2) For a fair comparison with the model output, I recommend you compute their transit time at the spin-up state, which better represent the equilibrium state of the model. In the current version, you compute model-derived transit times from a multi-year average, but this corresponds to a transient state where transit times are not unique.

Response: Thanks for the great suggestion. We will additionally analyze the modeled transit time over 1850-1860, when the models were spun up to steady state. Using the modeled data over 1850-1860 and 1995-2005 (the original results) both have pros and cons. For example, the estimated C transit time based on 1850-1860 results holds the equilibrium assumption but neglect the changes of C transit time over time (i.e., the observations are from the recent decades). The original results (i.e., over 1995-2005) were not derived from the equilibrium state, but they catch the time period of the observations. We will discuss this issue in the revised version.

3) Another aspect that requires clarification is the computation of the transit time distributions in Figure 1b. How were these distributions obtained from the data? Did you assume a specific distribution function and fitted its parameter values using the data? This seems to be the case for the 13 and the stock/flux data, but not for the incubations. Please clarify.

Response: We used the Gaussian kernel density estimation (KDE) to obtain the distributions in the Fig. 1b. We added the information in supplementary material.

References:

Carvalhais, N., Forkel, M., Khomik, M., Bellarby, J., Jung, M., Migliavacca, M., Mu, M., Saatchi, S., Santoro, M., Thurner, M., et al. (2014). Global covariation of carbon turnover times with climate in terrestrial ecosystems. Nature, 514(7521):213.

Lu, X., Wang, Y.-P., Luo, Y., and Jiang, L. (2018). Ecosystem carbon transit versus turnover times in response to climate warming and rising atmospheric CO2 concentration. Biogeosciences Discussions, 2018:1–22.

Rasmussen, M., Hastings, A., Smith, M. J., Agusto, F. B., Chen-Charpentier, B. M., Hoffman, F. M., Jiang, J., Todd-Brown, K. E. O., Wang, Y., Wang, Y.-P., and Luo, Y. (2016). Transit times and mean ages for nonautonomous and autonomous compartmental systems. Journal of Mathematical Biology, 73(6):1379–1398.

Sierra, C. A., Müller, M., Metzler, H., Manzoni, S., and Trumbore, S. E. (2017). The muddle of ages, turnover, transit, and residence times in the carbon cycle. Global Change Biology, 23(5):1763–1773.

References from author:

Bolker, B.M., Pacala, S.W. & Parton Jr, W.J. (1998) Linear analysis of soil decomposition: insights from the century model. Ecological Applications, 8, 425-439.

Liang, J., Li, D., Shi, Z., Tiedje, J.M., Zhou, J., Schuur, E.A.G., Konstantinidis, K.T. & Luo, Y. (2015) Methods for estimating temperature sensitivity of soil organic matter based on incubation data: A comparative evaluation. Soil Biology and Biochemistry, 80, 127-135.

Rasmussen, M., Hastings, A., Smith, M.J., Agusto, F.B., Chen-Charpentier, B.M., Hoffman, F.M., Jiang, J., Todd-Brown, K.E., Wang, Y., Wang, Y.P. & Luo, Y. (2016) Transit times and mean ages for nonautonomous and autonomous compartmental systems. J Math Biol, 73, 1379-1398.

Saoudi, S., Ghorbel, F. & Hillion, A. (1997) Some statistical properties of the kernel‐diffeomorphism estimator. Applied stochastic models and data analysis, 13, 39-58.

Sheather, S.J. & Marron, J.S. (1990) Kernel quantile estimators. Journal of the American Statistical Association, 85, 410-416.

Xu, X., Shi, Z., Li, D., Rey, A., Ruan, H., Craine, J.M., Liang, J., Zhou, J. & Luo, Y. (2016) Soil properties control decomposition of soil organic carbon: Results from data-assimilation analysis. Geoderma, 262, 235-242.

[Figure]

Please also note the supplement to this comment:
https://www.biogeosciences-discuss.net/bg-2018-342/bg-2018-342-AC2-supplement.pdf

————————————————————

[Figure]

**Fig. 1.** The diagram of model.

[Figure]

**Fig. 2.** The transit time calculated with CMIP5, fitted three-feedback-pools from pool-flux approach, and the whole observation transit time.

**Supplement:**

**Response text to the comments 1) and 2):**

*Model description*

In this study, a three-feedback-pool C model consisted with fast, slow, and a passive pools and carbon transfers between three pools. The dynamics of soil carbon pool follows first-order differential equation as described in the CENTURY and the Terrestrial Ecosystem models (Bolker *et al.*, 1998; Liang *et al.*, 2015). The pools and fluxes are shown in the following diagram:

[Figure]

The dynamics of the three C pools can be mathematically described as:

$$\begin{cases} \frac{dC_1(t)}{dt} = C_1(t\text{-}1)\text{-}Q10_1 \cdot k_1 \cdot C_1(t) + Q10_2 \cdot k_2 \cdot C_2(t) \cdot f_{12} + Q10_3 \cdot k_3 \cdot C_3(t) \cdot f_{13} \\ \frac{dC_2(t)}{dt} = C_2(t\text{-}1)\text{-}Q10_2 \cdot k_2 \cdot C_2(t) + Q10_1 \cdot k_1 \cdot f_{21} \\ \frac{dC_3(t)}{dt} = C_3(t\text{-}1)\text{-}Q10_3 \cdot k_3 \cdot C_3(t) + Q10_1 \cdot k_1 \cdot C_1(t) \cdot f_{31} + Q10_2 \cdot k_2 \cdot C_2(t) \cdot f_{32} \end{cases} \qquad (1)$$

where $C_1$ is used to describe soil carbon pool size, $f_{ij}$ is carbon transfer coefficient which indicating the fraction of the carbon entering i-th pool from j-th pool. $k_1$, $k_2$, $k_3$ is the decomposition rate in fast, slow and passive pools, respectively. carbon pool. $Q10_1$, $Q10_2$, $Q10_3$ is the temperature scalar in fast, slow and passive pools. For pool-flux approach, the in-site observation carbon dioxide in the three-pool model is the total carbon pool size and respiration. At steady state, soil respiration equals to carbon input in different biomes. Values in parenthesis indicated 99% confidence interval for predicted transit times at each site based on 50000 Monte Carlo calculations.

*Bayesian inversion with Markov Chain Monte Carlo approach*

The parameters in the three-pool model were estimated based on Bayes' theorem with the posterior probability density function of model parameters ($\theta$) (Liang *et al.*, 2015; Xu *et al.*,

2016). The prior knowledge of parameter represented by a prior probability density function P(θ) and the information in the pool-flux in-situ observational data represented by a likelihood function, P(Z|θ). The posterior probability density function of model can be described as equation (2):

$$P(\theta|Z) = \frac{P(\theta|Z) \cdot P(\theta)}{P(Z)} \tag{2}$$

In this study, we adopted the prior ranges of model parameter from Liang et al. (2015) across biomes (please see Table 1).

**Estimates of transit time and age from three-pool models**

Based on the concepts of mean age and mean transit time published by Rasmussen et al., (2016) (Rasmussen *et al.*, 2016), the mean carbon age defined as the whole time period when a carbon atom was respired into atmosphere from the entrance at a certain time, and then the mean age of carbon $\bar{a}_i(t)$ in a certain carbon pool i could be calculated with equation (3):

$$\bar{a}_i(t) = 1 + \frac{\sum_{i=1}^{3} (\bar{a}_j(t) - \bar{a}_j(t)) \cdot f_{ij}(t) \cdot C_i - \bar{a}_j(t) \cdot I_i(t)}{C_i} \tag{3}$$

where the $f_{ij}(t)$ is the carbon fraction transfer from j-th to i-th pools, $I_i(t)$ is the external input into this carbon pool.

The transit time $\tau_i(t)$ was defined as the mean age of carbon atoms leaving the carbon pool at a specific time. that is

$$\tau_i(t) = \sum_{i=1}^{d} f_i(t) \cdot a_i(t) \tag{4}$$

where the $f_i(t)$ is the fraction of carbon with mean age $a_i(t)$.

Table S1. Prior parameters of three-pool for the average of sites

| Parameter | Definition | Value | Range |
|---|---|---|---|
| Q10 | The temperature scalar in fast, slow and passive carbon pools | 2 | 0-6 |
| f12 | The fraction of carbon from pool 2 to pool 1 | 0.1 | 0.1-0.6 |
| f13 | The fraction of carbon from pool 3 to pool 1 | 0.2 | 0-1 |
| f21 | The fraction of carbon from pool 1 to pool 2 | 0.5 | 0.1-0.6 |
| f31 | The fraction of carbon from pool 1 to pool 3 | 0.004 | 0-0.1 |
| f32 | The fraction of carbon from pool 2 to pool 3 | 0.03 | 0-0.03 |
| k1 | The decomposition rate of the fast soil carbon pool | 0.01 | 0.001-0.05 |
| k2 | The decomposition rate of the slow soil carbon pool | 0.006 | 0.001-0.0021 |
| k3 | The decomposition rate of the passive soil carbon pool | 0.00002 | $1.9 \cdot 10^{-6}$-$2.1 \cdot 10^{-5}$ |

Table S2. Estimates of and parameter of soil carbon transit time across biomes.

| Parameter | | Boreal forest | Temp-erate forest | Tropi-cal forest | Crop-land | Tundra | Desert/ shrubl-and | Grassland/sa-vanna |
|---|---|---|---|---|---|---|---|---|
| $Q_{10}$ | C1 | 1.4 | 2.2 | 2.5 | 2.3 | 2.9 | 2.5 | 1.9 |
| | C2 | 2.8 | 1.4 | 1.1 | 1.3 | 4.2 | 1.3 | 1.1 |
| | C3 | 3.1 | 0.8 | 1.4 | 1.6 | 3.8 | 3.7 | 2.8 |
| Transit time (yr) | Tau 1 | 4.7 | 3.2 | 3.0 | 3.2 | 47.1 | 32.7 | 22.6 |
| | Tau 2 | 84.2 | 28.8 | 18.7 | 34.5 | 54.9 | 55.8 | 45.9 |
| | Tau 3 | 131.8 | 36.8 | 18.9 | 71.1 | 105.8 | 114.8 | 88.3 |
| | Mean | 66.4 | 79.0 | 28.9 | 77.1 | 166.5 | 135.3 | 53.8 |
| Mean age | | 1661.1 | 2761.5 | 366.5 | 1056.8 | 1805.7 | 1976.8 | 1127.4 |

Response text for comments 3): The KDE is a non-parametric approach to estimate the probability density function of a random variable. Let $(x_1, x_2, \cdots, x_n)$ denote the observed C transit time with density function $f$ as below:

$$\widehat{f}_h(x) = \frac{1}{nh} \sum_{i=1}^{n} K(\frac{x - x_i}{h})$$

where K is the non-negative function than integrates to one and has mean zero, and $h > 0$ is a smoothing parameter called the bandwidth. The density function was estimated by Gaussian Kernel Density (Sheather & Marron, 1990; Saoudi *et al.*, 1997), and the bandwidth for approaches of stable isotope $^{13}$C, pool-flux and incubation are: 48.61, 35.13, 2.62, respectively. We further show the probability density distribution of transit time among 12 models which shown as figure below.

[Figure]

Figure S1. The probability density function of CMIP5 output, the black line is the ensemble-model mean transit time.

[Figure]

FigureS2. Probability distributions of the parameters in the three -pool-feedback model for tundra ecosystem (See Equation (1) for abbreviations).

[Figure]

FigureS3. Probability distributions of the parameters in the three -pool-feedback model for boreal forest ecosystem (See Equation (1) for abbreviations).

[Figure]

FigureS4. Probability distributions of the parameters in the three -pool-feedback model for temperate forest (See Equation (1) for abbreviations).

[Figure]

FigureS5. Probability distributions of the parameters in the three-pool-feedback model for tropical forest (See Equation (1) for abbreviations).

[Figure]

FigureS6. Probability distributions of the parameters in the three -pool-feedback model for cropland (See Equation (1) for abbreviations).

---

## Author Response (AR1)

Dear Editor:

We are grateful for the reviewers and your efforts and time to improve our manuscript. In this version, we have made revisions as suggested by you and the two reviewers. Please find our materials in sequence as follow:

(1) We have answered the new three questions from the reviewers in the **Text 1**;

(2) We have strengthened the coherence of this article and made it more readable. The detailed information about the three-pool model has been provided in the main text as section 2.3 (*Methods*, 618 words) and 3.3 (*Results*, 193 words).

(3) The former response letter has been updated with the new results and is shown in **Text 2**.

**Text 1**

**Comment 1C:** *Please justify your use of the entirely new three-pool model, when there are a range of existing Community Land Models available?*

**Response:** The authors thank the reviewer for this suggestion. We have explained the reasons for selecting this three-pool model in the *Method* section of this revised version. We also have highlighted the existence of multiple model structures and recommend a biogeochemical model database in the discussion of the methodological uncertainty of this study. Please find our revisions in detail as below:

*The selection of three-pool model*: As mentioned by the reviewer, there is a range of existing Community Land Models (CLM) available. The Fig. R1.1 shows the structures of soil carbon model in the CENTURY model and the two recent versions of CLM. The latest version CLM (i.e., CLM5.0) used the same three-pool structure as CENTURY (panel a) and CLM4.5 (panel c). Many other global land models adopted the structure of CENTURY model due to its success of capturing the soil carbon dynamics over inter-annual to decadal timescales (Parton *et al.*, 1993; Luo *et al.*, 2015). Thus, we selected this three-pool model structure in our analysis (please see more on our reply to "Comment 1B").

[Figure]

Figure R1.1 The soil carbon model structures of the CENTRY, CLM-cn and CLM4.5 models.

*Highlight of the uncertainty from model structure:* Multicompartment models have been widely used to study soil carbon dynamics in current Earth system models (Manzoni and Porporato 2009; Sierra et al., 2015, 2018). For example, the terrestrial component of the HadGEM2 model (i.e., JUELS) uses a four-pool model to evaluate the soil organic (Collins et al., 2011). The MPI-ESM model represents the soil carbon stocks as two compartments (Roeckner et al., 2011). As shown in the Fig. R1.1, the CLM4-cn and CLM4.5 models differ in their structures of soil carbon component, but they both represent the soil organic carbon pool as multiple compartments (Lawrence et al., 2011; Oleson et al., 2013). The various structures of these soil carbon-cycle models have been nicely summarized into a "biogeochemical model database (bgc_md)" by the Theoretical Ecosystem Ecology Group in Max Planck Institute for Biogeochemistry (https://github.com/MPIBGC-TEE/bgc-md). Such effort is very important for evaluating the structure uncertainty in the estimates of soil C transit time. In this version, we have highlighted this point and cited the "bgc_md" as a supplementary for the readers (Line 19-24, Page 12):

> "As synthesized by Sierra et al. (2017), the observations of $\tau_{soil}$ are useful for a specific model once its pool structure is identified. This study also detect difference in the estimated $\tau_{soil}$ between the one- and three-pool models (Fig. 4). Thus, model database, such as the bgc-md (https://github.com/MPIBGC-TEE/bgc-md), could be a useful tool to improve the integration of observations and soil C models."

**Comment 2C:** *In your response to reviewer 2, please provide more detail on the new approach. What would drive soil C input? Would the model be calibrated against data from all sites?*

**Response:** Thanks. We have revised our responses to reviewer 2 with more detail on the new approach in this version. In brief, we assumed the total soil carbon input equals to total soil respiration at the steady state (Line 12-13, Page 6). The total C stocks and $CO_2$ efflux from observations were separated into pool-specific decomposition rates by deconvolution analysis (Fig.R1.1a, Liang et al., 2015). Although there are some parameters could not well-constrained, all the model would be calibrated against data (Table R1.1 and Fig. R2). The transit time of each biome were simulated with the equation (7) in our reply to Comment 2B.

Table R1.1 Maximum likelihod estimates of parameters, *P*-value, $R^2$ and the Akaike information criterion (*AIC*) values in the three-pool model with observations.

| Biomes | $Q_{10}$ | | | Transit time (year) | | | | *P* | $R^2$ | *AIC* |
|---|---|---|---|---|---|---|---|---|---|---|
| | fast | slow | passive | fast | slow | passive | Mean | | | |
| Boreal forest | 1.4 | 2.8 | 3.1 | 4.7 | 84.2 | 131.8 | 66.4 | < 0.05 | 0.95 | -158.9 |
| Temperate forest | 2.2 | 1.4 | 0.8 | 3.2 | 28.8 | 36.8 | 79 | < 0.05 | 0.96 | -167.5 |
| Tropical forest | 2.5 | 1.1 | 1.4 | 3 | 18.7 | 18.9 | 28.9 | < 0.05 | 0.95 | -224.7 |
| Cropland | 2.3 | 1.3 | 1.6 | 3.2 | 34.5 | 71.1 | 77.1 | < 0.05 | 0.99 | -209.5 |
| Tundra | 2.9 | 4.2 | 3.8 | 47.1 | 54.9 | 105.8 | 166.5 | < 0.05 | 0.96 | -106.1 |
| Desert/Shrubland | 2.5 | 1.3 | 3.7 | 32.7 | 55.8 | 114.8 | 135.3 | < 0.05 | 0.95 | -88.5 |
| Grassland/Savanna | 1.9 | 1.1 | 2.8 | 22.6 | 45.9 | 88.3 | 53.8 | < 0.05 | 0.95 | -45.8 |

In total, five out of eleven parameters were constrained for tundra ecosystem (Fig. S1). Eight out of eleven parameters were constrained for temperate, boreal forest and desert & shrubland (Fig. S2, S3, S6). Seven out of eleven parameters were constrained for tropical forest and cropland (Fig. S4, S5). For grassland and savanna, seven out of eleven parameters were constrained (Fig. S7).

[Figure]

Figure R1.2. The histogram of SOC transit time ($\tau_{soil}$) from observational data and simulation based on data assimilation analysis. The dash line indicates the median of the $\tau_{soil}$.

**Comment 3C:** *It is difficult to follow how transit time from the same Q10-based first-order kinetics model would be so insensitive to temperature (see Fig R2.1 in the supplement to their replies to my comments).*

**Response:** Sorry for confusion. The Fig. R2.1 shown the SOC $\tau_{soil}$ based on the observational data and CLM5 models results at the steady state, respectively. The transit times of 12 CMIP5 models were calculated by averaging $\tau_{soil}$ of each model. For each model, the $\tau_{soil}$ divided the SOC and NPP. Thus, the small SOC and NPP generated the small $\tau_{soil}$ in this study.

**Responses to Reviewer A**

**Comment 1A:** *In this paper, the authors evaluate soil carbon transit times in 12 CMIP5 models. They found that, compared to in-situ observations, transit times are usually underestimated by models, especially in cold regions and dry/hot regions. The authors show that some of these biases can be resolved by adopting more vertically-resolved parameterization of soil C dynamics with the CLM4.5 model.*

**Response:** Thanks for the clear summary of our manuscript.

**Comment 2A:** *I have concerns about this manuscript as it seems very similar to previous papers by e.g. Todd-Brown et al. (2013): the same models are evaluated with the same HWSD-MODIS based product. The novelty here is the comparison of models against transit times measured in worldwide soils, and I think it should be the main aim of the study. If the authors decide to keep the global evaluation, the HWSD-MODIS product should be confronted to in situ observations to justify its use as a global benchmark or, alternatively, the creation of this database could be used to derive a more robust global product.*

**Response:** We agree that Todd-Brown *et al.*, (2013) was the first study which has done the wonderful evaluation on the large uncertainty of soil C turnover time based on the HWSD-MODIS products and 12 CMIPS models. As pointed out by the reviewer, the unique contribution of our study is using the *in-situ* observations to benchmark the global models. In order to avoid the confusion, we have removed the results based on HWSD-MODIS products (i.e., the original panels c and d in the Fig. 3) in the revised version. Please see the updated Fig. 3 as below:

[Figure]

Figure R 2.1 Relationships between SOC transit time ($\tau_{soil}$) and climate factors in both observations and CIMP5 models. The black solid lines show the negative correlation between $\tau_{soil}$ and (**a**) mean annual temperature and (**b**) mean annual precipitation. The black dots indicate the aggregated $\tau_{soil}$ over each category of MAT ($y$= -5.47$x$+1971.5, $r^2$ = 0.49, *P*<0.01) or MAP ($y$= -68.19$x$+1222.6, $r^2$ = 0.60, *P*<0.01). The red and blue dots

present the mean value of the multiple models based on the ratios of carbon stock over NPP and $R_h$, respectively.

**Comment 3A:** *Section 3.2 is very hard to understand. It is not clear whether models are evaluated against the in-situ observations, or whether they are evaluated against the HWSDMODIS based global product (as it seems in Fig. 3). The discussion around improvements due to the addition of a vertical resolution in CLM4.5 is reduced to less than 10 lines while it seems to be one of the key findings of the whole study.*

**Response**: The Section 3.2 is mainly the evaluation of models against the *in-situ* observations. In this version, we have made this section clearer by:

(1) We have added more details about the comparison between model results and the in-situ observations. In brief, only the grid cells containing the locations of in-situ observations were selected from the models for the comparison.
(2) To avoid confusion, we have removed the HWSD/MODIS results in this version.
(3) The results based on the vertical resolution in CLM4.5 have be expanded in the section 3.4.

*Specific Comments:*

**Comment 4A:** *p3 l 21-29: which period of the historical simulation did the authors consider?*

**Response**: The historical period is from 1850 to 1860. We have made it clear in the revision.

**Comment 5A:** *p3 l30: I find that there is a missed opportunity here to use in situ observations to derive a more robust global dataset of transit times. HWSD and MODIS NPP both come with known biases and there may be other products to choose from e.g. soilgrids (www.soilgrids.org).*

**Response**: As mentioned above, we will remove the results based on HWSD and MODIS in the revised version. We thank the reviewer for the suggestion of deriving a robust global dataset of transit time based on the observations. This task is scientifically very important, but is difficult at the current stage due to a few reasons. First, the available observations are limited by the unequal quality and the uneven spatial distribution of the locations. Second, no data-driven approach is ready for deriving a global dataset of C transit time based on the observations. Third, it is difficult to reduce the methodological uncertainty of data (e.g., Fig. 1b) in integrating them into a given model for global calculation. We will discuss this issue in the revised manuscript.

**Comment 6A:** *p6 l30-35: I do not understand what is learned by replacing MODIS NPP with TRENDY models (which ones? reference is missing here). Does that mean that TRENDY is considered as an observation of NPP against which ESMs are evaluated?*

**Response**: The results from TRENDY and MODIS NPP have been removed in this version. Also, we agree with the reviewer that TRENDY NPP cannot used as observations.

**Comment 7A** *Figure 2: from the legend, panels c and d are missing. Panel a is hard to understand and uncertainties are missing from panel b.*

**Response**: Sorry for the confusion. We have corrected the figure legend in the revised version. More sentences will be added to explain the panel a, and the uncertainties have been added in the panel b.

**Comment 8A** *Figure 3: in panel a and b, do black dots represent data from the 187 sites? or were they extracted from the HWSD/MODIS product?*

**Response**: The black dots represent data from 187 sites in panel (a) and (b) in Fig. 3, we grouped them into different levels of climatic variables. We will revise the figure legend to make it clearer. Also, the panels *c* and *d* will be removed to avoid confusion.

**Review B:**

**Comment 1B:** *In this manuscript, the authors present observation-based estimates of transit times of carbon in soils and compare these estimates with model predictions. This is an important topic because transit times are a very good constraint for evaluating model performance. There has been a lot of recent research on this topic, motivated in part by the work of Carvalhais et al. (2014), who used a stock-over-flux approach to compute residence times from models and observations. Recent publications have shown that this approach has problems to compute transit times for systems of multiple pools and out of equilibrium (Lu et al., 2018; Sierra et al., 2017), and better methods for estimating transit times for systems out of equilibrium have been developed (e.g. Rasmussen et al., 2016).*

**Response**: We thank the reviewer for the great summary. We have revise the introduction of our manuscript to highlight these milestone works as mentioned by the reviewer.

*Major remark:*
**Comment 2B**: *Despite these recent developments, this manuscript uses observations from incubation experiments and $^{13}C$ measurements from $C_3/C_4$ vegetation replacement experiments, in which the rate of soil carbon loss is estimated assuming one single pool in equilibrium. This is evidenced by equations (1) to (3) in Text S1 of the supplementary material. The implication of this assumption is that the observations are treated as a homogeneous system, without differentiating between the age of the stored carbon and the age of the carbon in the efflux. In the introductory paragraphs, the manuscript gives the impression that it provides an advance by providing observation estimates of transit times, but in reality, these estimates suffer the same problems of previous approaches.*

*I recommend the authors to use the data they compiled to fit multiple-pool models to better estimate age and transit times from these observations. You probably would still need to keep the steady-state assumption for this type of observations, but at least you can remove the one-single homogeneous pool assumption.*

**Response**: Many thanks for thoughtful suggestion. We have followed the reviewer's suggestion to fit the data to a three-pool model in addition to the single pool approach. Then, we estimated the C transit time from the observations under the steady-state assumption. The reasons for the selection of the three-pool model has been summarized in our above reply to the "Comment 1C". As shown in the Fig. R2.3, there are differences in the estimated $\tau_{soil}$ between the one- and three-pool models. However, the underestimation of $\tau_{soil}$ by the CMIP5

models were detected by both of one- and three-pool models. The estimated parameters and new results could be found in the Table R1.1and Table 2.1. Please also find the details of the three-pool model as below:

The dynamics of SOC are widely represented by models with multiple pools. For the better estimation of the carbon transit times, we fitted a three-pool model with the observational data derived from stock-over-flux approach. In this study, a three-pool C model consisted of fast, slow, and passive pools and carbon transfers among three pools (Fig. R2.2). This model shares the same framework with the CENTURY and the CLM4.5 (Fig. R1.1). The dynamics of soil carbon pools follow the first-order differential kinetics. Based on the theoretical analysis, the C dynamics of the three-pool model can be mathematically described by the following matrix equation (Luo *et al.*, 2003; Xia *et al.*, 2013) as:

$$\frac{dC(t)}{dt} = I(t) - AKC(t) \tag{1}$$

[revised manuscript text omitted]

**Comment 3B:** *For a fair comparison with the model output, I recommend you compute their transit time at the spin-up state, which better represent the equilibrium state of the model. In the current version, you compute model-derived transit times from a multi-year average, but this corresponds to a transient state where transit times are not unique.*

**Response**: Thanks for the great suggestion. We have additionally analyzed the modeled C transit time over 1850-1860, when the models were close to the steady state. Using the modeled data over 1850-1860 and 1995-2005 (the original results) both have pros and cons. For example, the estimated C transit time based on 1850-1860 results holds the equilibrium assumption but neglect the changes of C transit time over time (i.e., the observations are from the recent decades). The original results (i.e., over 1995-2005) were not derived from the equilibrium state, but they catch the time period of the observations. However, the results were consistent by using the modelled data over 1850-1860 and 1995-2005.

**Comment 4B:** *Another aspect that requires clarification is the computation of the transit time distributions in Figure 1b. How were these distributions obtained from the data? Did you assume a specific distribution function and fitted its parameter values using the data? This seems to be the case for the 13 and the stock/flux data, but not for the incubations. Please clarify.*

**Response**: The Gaussian kernel density estimation (KDE) was used to obtain the distributions in the Figure 1b. The $^{13}$C, stock/flux data and incubations both are follow the Gaussian kernel density distributions. The $\tau_{soil}$ from laboratory studies was significantly shorter than the other two methods, therefore they shared the different axis in Fig. 1b. The left axis is for $^{13}$C and *stock-over-flux* approaches, and the right axis is for laboratory incubation studies. We have added illustration distribution function and detailed information in section 2.4 (*Statistical analysis*, Line 3-11, Page 10) as follow:

The Gaussian kernel density estimation (KDE) was used to obtain the distributions of observed transit times (Sheather & Marron, 1990; Saoudi *et al.*, 1997). The KDE is a non-parametric approach to estimate the probability density function of a random variable. Let $(x_1, x_2, \cdots, x_n)$ denote the observed SOC $\tau_{soil}$ with density function f as below:

$$\hat{f}_h(x) = \frac{1}{nh} \sum_{i=1}^{n} K\left(\frac{x - x_i}{h}\right) \tag{8}$$

where K is the non-negative density function than integrates to one and has mean zero, and $h > 0$ is a smoothing parameter called the bandwidth. The bandwidth for approaches of stable isotope $^{13}$C, *stock-over-flux* and incubation are: 48.61, 35.13, 2.62, respectively.

**References from reviewer A:**

Todd-Brown, K.E., Randerson, J.T., Post, W.M., Hoffman, F.M., Tarnocai, C., Schuur, E.A. and Allison, S.D.: Causes of variation in soil carbon simulations from CMIP5 Earth system models and comparison with observations. Biogeosciences, 10, 1717-1736, https://doi: 10.5194/bg-10-1717-2013.

**References from reviewer B:**

Carvalhais, N., Forkel, M., Khomik, M., Bellarby, J., Jung, M., Migliavacca, M., Mu, M., Saatchi, S., Santoro, M., Thurner, M., et al. (2014). Global covariation of carbon turnover times with climate in terrestrial ecosystems. Nature, 514(7521):213.

Lu, X., Wang, Y.-P., Luo, Y., and Jiang, L. (2018). Ecosystem carbon transit versus turnover times in response to climate warming and rising atmospheric CO2 concentration. Biogeosciences Discussions, 2018:1–22.

Rasmussen, M., Hastings, A., Smith, M. J., Agusto, F. B., Chen-Charpentier, B. M., Hoffman, F. M., Jiang, J., Todd-Brown, K. E. O., Wang, Y., Wang, Y.-P., and Luo, Y. (2016). Transit times and mean ages for nonautonomous and autonomous compartmental systems. Journal of Mathematical Biology, 73(6):1379–1398.

Sierra, C. A., Müller, M., Metzler, H., Manzoni, S., and Trumbore, S. E. (2017). The muddle of ages, turnover, transit, and residence times in the carbon cycle. Global Change Biology, 23(5):1763–1773.

**References from author**

Bolker, B.M., Pacala, S.W. & Parton Jr, W.J. (1998) Linear analysis of soil decomposition: insights from the century model. *Ecological Applications*, **8**, 425-439.

Collins, W. J., Bellouin, N., Doutriaux-Boucher, M., Gedney, N., Halloran, P., Hinton, T., ... & Martin, G. (2011). Development and evaluation of an Earth-System model–HadGEM2. Geoscientific Model Development, 4(4), 1051-1075.

Lawrence, D. M., Oleson, K. W., Flanner, M. G., Thornton, P. E., Swenson, S. C., Lawrence, P. J., ... & Bonan, G. B. (2011). Parameterization improvements and functional and structural advances in version 4 of the Community Land Model. Journal of Advances in Modeling Earth Systems, 3(1).

Liang, J., Li, D., Shi, Z., Tiedje, J.M., Zhou, J., Schuur, E.A.G., Konstantinidis, K.T. & Luo, Y. (2015) Methods for estimating temperature sensitivity of soil organic matter based on incubation data: A comparative evaluation. *Soil Biology and Biochemistry*, **80**, 127-135.

Luo, Y., Gerten, D., Le Maire, G., Parton, W. J., Weng, E., Zhou, X., ... & Dukes, J. S. (2008). Modeled interactive effects of precipitation, temperature, and [CO2] on ecosystem carbon and water dynamics in different climatic zones. Global Change Biology, 14(9), 1986-1999.

Luo, Y., & Weng, E. (2011). Dynamic disequilibrium of the terrestrial carbon cycle under global change. Trends in Ecology & Evolution, 26(2), 96-104.

Luo, Y., Shi, Z., Lu, X., Xia, J., Liang, J., Jiang, J., ... & Chen, B. (2017). Transient dynamics of terrestrial carbon storage: mathematical foundation and its applications.

Manzoni, S., & Porporato, A. (2009). Soil carbon and nitrogen mineralization: theory and models across scales. Soil Biology and Biochemistry, 41(7), 1355-1379.

Oleson, K. W., Lawrence, D. M., Gordon, B., Flanner, M. G., Kluzek, E., Peter, J., ... & Heald, C. L. (2010). Technical description of version 4.0 of the Community Land Model (CLM).

Parton, W. J., Schimel, D. S., Cole, C. V., & Ojima, D. S. (1987). Analysis of factors controlling soil organic matter levels in Great Plains Grasslands 1. Soil Science Society of America Journal, 51(5), 1173-1179.

Rasmussen, M., Hastings, A., Smith, M.J., Agusto, F.B., Chen-Charpentier, B.M., Hoffman, F.M., Jiang, J., Todd-Brown, K.E., Wang, Y., Wang, Y.P. & Luo, Y. (2016) Transit times and mean ages for nonautonomous and autonomous compartmental systems. *J Math Biol*, **73**, 1379-1398.

Roeckner, E., Giorgetta, M. A., Crueger, T., Esch, M., & Pongratz, J. (2011). Historical and future anthropogenic emission pathways derived from coupled climate–carbon cycle simulations. *Climatic Change*, **105**, 91-108.

Saoudi, S., Ghorbel, F. & Hillion, A. (1997) Some statistical properties of the kernel‐diffeomorphism estimator. *Applied stochastic models and data analysis*, **13**, 39-58.

Sheather, S.J. & Marron, J.S. (1990) Kernel quantile estimators. *Journal of the American Statistical Association*, **85**, 410-416.

Xu, X., Shi, Z., Li, D., Rey, A., Ruan, H., Craine, J.M., Liang, J., Zhou, J. & Luo, Y. (2016) Soil properties control decomposition of soil organic carbon: Results from data-assimilation analysis. *Geoderma*, **262**, 235-242.

Sierra, C. A., Harmon, M. E., & Perakis, S. S. (2011). Decomposition of heterogeneous organic matter and its long‐term stabilization in soils. Ecological Monographs, **81**, 619-634.

Sierra, C. A., & Müller, M. (2015). A general mathematical framework for representing soil organic matter dynamics. Ecological Monographs, **85**, 505-524.

**Supplementary figures:**

[Figure]

Figure S1. Probability distributions of the parameters in the three-pool model for tundra ecosystem (See Equation (1) for abbreviations).

[Figure]

Figure S2. Probability distributions of the parameters in the three-pool model for boreal forest ecosystem (See Equation (1) for abbreviations).

[Figure]

Figure S3. Probability distributions of the parameters in the three-pool model for temperate forest (See Equation (1) for abbreviations).

[Figure]

Figure S4. Probability distributions of the parameters in the three-pool model for tropical forest (See Equation (1) for abbreviations).

[Figure]

Figure S5. Probability distributions of the parameters in the three-pool model for cropland (See Equation (1) for abbreviations).

[Figure]

Figure S6. Probability distributions of the parameters in the three-pool model for desert & shrubland (See Equation (1) for abbreviations).

[Figure]

Figure S7. Probability distributions of the parameters in the three-pool model for grassland & savanna (See Equation (1) for abbreviations).

---

## Author Response (AR2)

Dear Dr. Subke,

We really appreciate your time and effort on improving our manuscript. We have revised these issues point by point as follow. Please note that the comments are in italic gray followed by our replies in black.

**Comment 1:** *Page. 2, Line. 2: Please capitalize first letter of all words in Earth System Models.*
**Response:** Done. the first letter of all words in Earth System Model has capitalized through the whole text.

**Comment 2:** *P. 2, L. 3: "mean age" of C atoms is not correct. Please change to "mean residence time".*
**Response:** Thanks for the suggestion. We have carefully double checked the differences in the concepts of transit time, residence time and mean age. The recent book *Radiocarbon and Climate Change* wrote by Schuur et al. (2016) has clearly summarized their difference. Below is the table on the Page 71 of that book:

**Table 3.2** Terms used to infer C dynamics using biogeochemical box models.

| Term | Concept |
| --- | --- |
| Turnover time | An inventory divided by either the input or the output flux. |
| Age | Time since a C atom in a system entered the it. Integrating over all atoms present in the system at a given time, one can calculated a mean age. |
| Transit time | Time it takes a C atom to move through (transit) the system, which is equivalent to the age of the atoms at the time they leave it. Integrating over all atoms in the output flux, one can calculated the mean transit time. |
| Residence time | Variably defined in the literature, sometimes equivalent to age, sometimes to transit time. The use of this term depends on the scientific discipline and the context where it is applied. |
| Conventional radiocarbon age | Time since C in a system was fixed from the atmosphere as determined by the radiocarbon dating method using Libby's half-life, and assuming a closed system for C. |
| Calibrated radiocarbon age | Calendar year in which the C in system was fixed from the atmosphere, assuming a closed system and using a stated calibration curve. |

Note that only in single-pool, homogeneous systems in steady-state (one for which the probability of every atom leaving is equal), are *turnover time*, *mean age*, *mean transit time*, and *mean residence time* equal.

It shows that the residence time is variably defined in different studies. Another recent study of Serria et al. (2016) also suggest the research community of soil carbon to avoid using the concept of "residence time". Thus, in this version, we kept the definition as same as that used in the book of Schuur et al. (2016) as:

> "… soil carbon (C) transit time ($\tau_{soil}$), which quantifies the age of the C atoms at the time they leave the soil."

**Comment 3:** *P. 2, L. 7: Please change: "(median of 4 years, with interquartile range of 1 to 25 years)".*

**Response:** Done. "(median of 4 years, with interquartile range of 1 to 25 years)" has revised as "(4 with 1 to 25 years)" in *Abstract* and *Results* section.

**Comment 4:** *P. 2, L 11-14: Make 2 sentences, with full stop after "temperature" (end of line 12).*

**Response:** This long sentence has revised to two sentences in *Abstract* section (P. 2, L. 11-13) as:

"We then found a significant and negative linear correlation between the *in-situ* measured $\tau_{soil}$ and mean annual air temperature. The underestimations of modeled $\tau_{soil}$ are mainly located in cold and dry biomes, especially tundra and desert."

**Comment 5:** *P. 2, L. 13: Comma needed after "biomes".*

**Response:** Done.

**Comment 6:** *P. 2, L. 17: Please change to: "... soil C dynamics in regions limited by temperature or moisture".*

**Response:** Done. The sentence has rephrased in *Abstract* section (P. 2, L. 15-17) as:

"These findings indicate that the spatial variation of $\tau_{soil}$ is a useful benchmark for ESMs, and we recommend more observations and modeling efforts on soil C dynamics in regions limited by temperature and moisture."

**Comment 7:** *P. 3, L. 3: As above, please use capitals for each of the words "Earth System Models".*

**Response:** Done.

**Comment 8:** *P. 3, L. 9-10: Please rephrase to: "It is difficult to reduce or even diagnose this uncertainty, as many processes…"*

**Response:** Done. This sentence has rephrased in *Introduction* section (P3, L 9-11) as:

"It is difficult to reduce or even diagnose this uncertainty, as many processes collectively affect the time of C atoms transit the soil system (i.e., transit time; $\tau_{soil}$)"

**Comment 9:** *P. 3, L. 19/20: Please rephrase: "…, and the construction of a benchmarking database of available observation is urgently needed (Koven et al., 2017).*

**Response:** Done. This sentence has rephrased in *Introduction* section (P. 3, L. 17-20) to:

"Therefore, identifying the locations of such underestimations is critical to improve the predictive ability of ESMs on terrestrial C cycle, and the construction of a benchmarking database of available observations is urgently needed (Koven et al., 2017)."

**Comment 10:** *P. 3, L. 24: Rephrase: "…commonly defined as "turnover time", calculated by dividing SOC stock by C fluxes such as …".*

**Response:** Done. This sentence has rephrased in *Introduction* section (P3, L 23-25) to:

"The first approach commonly defined as "*turnover time*", calculated by the division of SOC stock by C fluxes such as net primary productivity (NPP) or heterotrophic respiration ($R_h$)."

**Comment 11:** P. 4, L. 5: *"mean residence times" rather than "mean ages".*

**Response:** Thanks for the suggestion. Please see **Comment 2**. we kept the definition as same as that used in the book of Schuur et al. (2016).

**Comment 12:** *P. 4, L. 20-22: This sentence is not clear. I suggest making it into 2 sentences: "$\tau_{soil}$ was calculated under the homogenous one-pool assumption at steady state for all studies. Data from observations and CMIP5 ensemble were then used to calculate the $\tau_{soil}$ based on both one-pool and three-pool models.*

**Response:** Done. This sentence has separated two sentences to made it clearer in *Introduction* section (P 4, L19-21) as:

> "The SOC $\tau_{soil}$ was calculated under the homogenous one-pool assumption at steady state for all studies. Data from observations and CMIP5 ensemble were then used to calculate the $\tau_{soil}$ based on both one-pool and three-pool models."

**Comment 13:** *P. 5, L. 4/5: Use active voice; "We constructed a database…".*
**Response:** Done.

**Comment 14:** *P. 10, L. 14: Text in brackets: "(median of 60 years, with interquartile range of 8 to 29 years)". For subsequent results, simply state "(X with Y to Z years)".*
**Response:** Done. The subsequent results were simply state as "(X with Y to Z years)" in the main text.

**Comment 15:** *P. 11, L. 11: adjust spaces: "0.5° x 0.5°" (i.e. spaces either side of x, but not before degree sign).*
*P. 11, L. 14: delete "was" at end of line*
*P. 12, L. 5: "show", not "shown". Delete "both" at end of line.*
*P. 12, L. 18: delete "the" before "equation (12)". Change "showed" to "show" at end of the line.*
*P. 12, L. 19: delete "the" before CLM4.5". Change "Fig. 6" to "Fig. 5".*
**Response:** Done.

**Comment 16:** *P. 12, L. 24: Rephrase: "Higher NPP values simulated by ESMs…"*
**Response:** Done. This sentence has rephrased in section 3.4 (P. 12, L.24-25) as

> "Higher NPP values simulated by ESMs in the cold and dry regions have been reported by previous studies (Shao et al., 2013, Smith et al., 2016, Xia et al., 2017)".

**Comment 17:** *P. 13, L.4-6: However, other processes such as the microbial dynamics, SOC stabilization and nutrient cycles could affect $\tau_{soil}$, but are so far not fully considered by CMIP5 models (Luo et al., 2016).*
**Response:** Done. This sentence has rephrased in section 3.4 (P. 13, L. 4-6) to

> "However, other processes such as the microbial dynamics, SOC stabilization and nutrient cycles could affect the estimation of $\tau_{soil}$, but are so far fully considered by the CMIP5 models (Luo et al., 2016)."

**Comment 18:** *P. 13, L. 23: Is it fair to say that the model data base "is" a useful tool (rather than "could be"?*
**Response:** Done. Yes, "…is a useful tool…" is better. This sentence has revised as

> "Thus, model database, such as the bgc-md (https://github.com/MPIBGC-TEE/bgc-md), is

a useful tool to improve the integration of observations and soil C models.".

**Comment 19:** *P. 13, L. 24: Delete "Thus".*
**Response:** Done.

**Reference:**
Schuur, E. A., Druffel, E. R., & Trumbore, S. E. (2016). *Radiocarbon and climate change*. Cham, Switzerland: Springer.
Sierra, C. A., Müller, M., Metzler, H., Manzoni, S., & Trumbore, S. E. (2016). The muddle of ages, turnover, transit, and residence times in the carbon cycle. *Global change biology*, *23*(5), 1763-1773.

---

## Author Response (AR3)

Dear Dr. Subke,

We really appreciate your help. We have double checked our manuscript and revised this issue. Please note that the comments are in italic gray followed by our replies in black.

**Comment 1**: *P. 4, l. 7: Apologies if this was not clear. For the first time when you state median in interquartile range, please make this clear. So in this line, please state "(Median = 4 years; interquartile range = 1 to 25 years)". Following that, please use "31; 5 to 84 years", etc. within the abstract. In the results section, please repeat the full text in brackets, i.e. on page 10, line 15, please state "(Median = 60 years; interquartile range = 8 to 29 years)". From then, simplify to "16; 3 to 156 years", etc.*

**Response:** Done. The median in interquartile range were used in *Abstract* and *Result* sections for the first time. The simplified transit times were used as "X; X to XX years" afterwards.